# LLM Blueprint: Enabling Text-to-Image Generation with Complex and Detailed Prompts

**Hanan Gani[1], Shariq Farooq Bhat[2], Muzammal Naseer[1], Salman Khan[1,3], Peter Wonka[2]**
[1]Mohamed Bin Zayed University of AI    [2]KAUST    [3]Australian National University
{hanan.ghani, muzammal.naseer, salman.khan}@mbzuai.ac.ae
shariq.bhat@kaust.edu.sa,    pwonka@gmail.com

| Text Prompt | object descriptions | spatial relationships |
|---|---|---|
| In a cozy living room, a heartwarming scene unfolds. A friendly and affectionate Golden Retriever with a soft, golden-furred coat rests contently on a plush rug, its warm eyes filled with joy. Nearby, a graceful and elegant white cat stretches leisurely, showcasing its pristine and fluffy fur. A sturdy wooden table with polished edges stands gracefully in the center, adorned with a vase of vibrant flowers adding a touch of freshness. On the wall, a sleek modern television stands ready to provide entertainment. The ambiance is warm, inviting, and filled with a sense of companionship and relaxation. | - *golden retriever:* resting on a plush rug with a soft golden fur coat
- *cat:* white, graceful and elegant with fluffy fur
- *table:* wooden
- *flower vase:* vibrant
- *television:* sleek modern
- *living room:* warm-ambience | - golden retriever and cat sitting nearby
- table stands at the center
- flower vase kept on the table
- television is fixed on the wall |

Figure 1: Current state-of-the-art text-to-image models (Columns 1-4) face challenges when dealing with lengthy and detailed text prompts, resulting in the exclusion of objects and fine-grained details. Our approach (Column 5) adeptly encompasses all the objects described, preserving their intricate features and spatial characteristics as outlined in the two white boxes.

## Abstract

Diffusion-based generative models have significantly advanced text-to-image generation but encounter challenges when processing lengthy and intricate text prompts describing complex scenes with multiple objects. While excelling in generating images from short, single-object descriptions, these models often struggle to faithfully capture all the nuanced details within longer and more elaborate textual inputs. In response, we present a novel approach leveraging Large Language Models (LLMs) to extract critical components from text prompts, including bounding box coordinates for foreground objects, detailed textual descriptions for individual objects, and a succinct background context. These components form the foundation of our layout-to-image generation model, which operates in two phases. The initial *Global Scene Generation* utilizes object layouts and background context to create an initial scene but often falls short in faithfully representing object characteristics as specified in the prompts. To address this limitation, we introduce an *Iterative Refinement Scheme* that iteratively evaluates and refines box-level content to align them with their textual descriptions, recomposing objects as needed to ensure consistency. Our evaluation on complex prompts featuring multiple objects demonstrates a substantial improvement in recall compared to baseline diffusion models. This is further validated by a user study, underscoring the efficacy of our approach in generating coherent and detailed scenes from intricate textual inputs. Our code is available at https://github.com/hananshafi/llmblueprint.

# 1 INTRODUCTION

Modern generative diffusion models, e.gRombach et al. (2022); Ho et al. (2020); Saharia et al. (2022); Ruiz et al. (2023), provided a massive leap forward in the problem of text-to-image generation and have emerged as powerful tools for creating diverse images and graphics from plain text prompts. Their success can be attributed to several factors, including the availability of internet-scale multi-modal datasets, increased computational resources, and the scaling up of model parameters. These models are trained using shorter prompts and especially excel at generating images of one prominent foreground object. However, as the description length and the number of objects in the scene increase, modern diffusion models tend to ignore parts of the prompt often leading to critical omissions, misrepresentations, or the generation of objects that do not align with the nuanced details described in the prompts. Fig. 1 shows a scenario where existing state-of-the-art text-to-image diffusion models struggle to follow all the details. This failure can partly be ascribed to the diffusion model's CLIP text encoder (Radford et al., 2021) which can only process the first 77 text tokens, effectively truncating longer prompts and potentially omitting critical details. Indeed, a single prompt describing a complex scene can span far beyond these token limits, making it a challenge for existing models to process and translate long prompts comprehensively.

Recent efforts (Epstein et al., 2023; Kang et al., 2023) have been dedicated to improving the capabilities of pre-trained diffusion models to faithfully follow the intricate details within text prompts. These works predominantly revolve around aspects such as object count (e.g. *"2 oranges and 4 apples on the table"*), and/or capturing spatial relationships among objects (e.g. *"an orange on the left of an apple"*). In the context of longer and more complex prompts, these models still tend to struggle to generate coherent images that faithfully reflect the complexity of the text prompts, especially when tasked with the placement of object instances at considerable spatial separations, often falling short of comprehensively capturing all instances of objects as intended. More recently layout-based diffusion models (Feng et al., 2023; Li et al., 2023; Yang et al., 2023b) have proven to be effective in capturing the count and spatial characteristics of the objects in the prompt. Such models first generate bounding boxes of all the objects and then condition the diffusion model jointly on the bounding boxes and the text prompt to generate the final image. While effective in the case of small prompts, these models still struggle when presented with long text descriptions that feature multiple diverse objects and hence fail to generate the desired output (See Fig. 1).

To address these challenges, our approach seeks to improve text-to-image generation from lengthy prompts. We introduce a framework that divides image generation into two phases: generating a global scene, followed by iterative refinement of individual object representations. We exploit LLMs to break down long prompts into smaller components organized in a data structure that we call *Scene Blueprint*. This allows us to generate the image in a step-wise manner. Our framework ensures that the final image faithfully adheres to the details specified in lengthy and complex text prompts.

We evaluate our framework on challenging prompts containing 3 to 10 unique foreground objects in varied scenes. Our results showcase a significant improvement in recall ($\sim$85%) compared to the baseline Feng et al. (2023) ($\sim$69%) (+16 % improvement). We also include a user study that demonstrates that our proposed method consistently produces coherent images that closely align with their respective textual descriptions, whereas existing approaches struggle to effectively handle longer text prompts (See Fig. 4)

In summary, our main contributions are:

- **Iterative image generation from long prompts**: We introduce a two-phase framework for generating images from long textual descriptions, ensuring a faithful representation of details.

- **Scene Blueprints using LLMs:** We propose *Scene Blueprints* as a structured scene representation encompassing scene layout and object descriptions that enable a coherent step-wise generation of images from complex and lengthy prompts.

- **State-of-the-art results:** We present quantitative and qualitative results showcasing the effectiveness of our method, in terms of adherence to textual descriptions, demonstrating its applicability and superiority in text-to-image synthesis from lengthy prompts.

## 2 RELATED WORK

**Text-to-Image Diffusion.** Over the years, Generative Adversarial Networks (GAN) (Goodfellow et al., 2014) have been the default choice for image synthesis (Brock et al., 2018; Reed et al., 2016; Xu et al., 2018; Zhang et al., 2017; 2021; Tao et al., 2022; Zhang et al., 2018a; Karras et al., 2019). However, more recently, the focus has shifted towards text conditioned autoregressive models (Ding et al., 2021; Gafni et al., 2022; Ramesh et al., 2021; Yu et al., 2022) and diffusion models (Rombach et al., 2022; Gu et al., 2022; Nichol et al., 2021; Ramesh et al., 2022; Saharia et al., 2022) which have exhibited impressive capabilities in producing high-quality images while avoiding the training challenges, such as instability and mode collapse, commonly associated with GANs (Arjovsky et al., 2017; Gulrajani et al., 2017; Kodali et al., 2017). In particular, diffusion models are trained on large-scale multi-modal data and are capable of generating high-resolution images conditioned on text input. Nevertheless, effectively conveying all the nuances of an image solely from a text prompt can present a considerable hurdle. Recent studies have demonstrated the effectiveness of classifier-free guidance (Ho & Salimans, 2022) in improving the faithfulness of the generations in relation to the input prompt. However, all these approaches are designed to accept shorter text prompts, but they tend to fail in scenarios where the prompt describing a scene is longer. In contrast, our proposed approach generates images from longer text prompts, offering an efficient solution to address this challenge.

**Layout-to-Image Generation.** Generating images from layouts either in the form of labeled bounding boxes or semantic maps was recently explored in (Sun & Wu, 2019; Sylvain et al., 2021; Yang et al., 2022; Fan et al., 2023; Zhao et al., 2019; Park et al., 2019). Critically, these layout to image generation methods are only conditioned on bounding boxes and are closed-set, i.e., they can only generate limited localized visual concepts observed in the training set. With the inception of large multi-modal foundational models such as CLIP (Radford et al., 2021), it has now been possible to generate images in an open-set fashion. Diffusion-based generative models can be conditioned on multiple inputs, however, they have been shown to struggle in following the exact object count and spatial locations in the text prompts (Chen et al., 2023; Kang et al., 2023). More recently layout conditioned diffusion models have been proposed to solve this problem (Chen et al., 2023; Li et al., 2023; Yang et al., 2023b; Phung et al., 2023). Chen et al. (2023) manipulates the cross-attention layers that the model uses to interface textual and visual information and steers the reconstruction in the desired user-specified layout. GLIGEN (Li et al., 2023) uses a gated self-attention layer that enables additional inputs (e.g., bounding boxes) to be processed. ReCo (Yang et al., 2023b) achieves layout control through regional tokens encoded as part of the text prompt. Zheng et al. (2023) introduce LayoutDiffusion which treats each patch of the image as a special object for multimodal fusion of layout and image and generates images with both high quality and diversity while maintaining precise control over the position and size of multiple objects. In addition to this, there have been few works on LLM-based layout generation (Feng et al., 2023; Lian et al., 2023). These works exploit the LLMs' abilities to reason over numerical and spatial concepts in text conditions (Li et al., 2022a). Building upon these works, we extend LLMs' powerful generalization and reasoning capabilities to extract layouts, background information, and foreground object descriptions from longer text prompts.

**Diffusion Based Image Editing and Composition.** Diffusion-based image editing has received overwhelming attention due to its ability to condition on multiple modalities. Recent works utilize text-based image editing using diffusion models to perform region modification. DiffusionCLIP (Kim et al., 2022) uses diffusion models for text-driven global multi-attribute image manipulation on varied domains. Liu et al. (2023) provides both text and semantic guidance for global image manipulation. In addition, GLIDE (Nichol et al., 2021) trains a diffusion model for text-to-image synthesis, as well as local image editing using text guidance. Image composition refers to a form of image manipulation where a foreground reference object is affixed onto a designated source image. A naive way to blend a foreground object on a background image may result in an unrealistic composition. However, more recent works (Avrahami et al., 2021; Yang et al., 2023a; Ho & Salimans, 2022; Lu et al., 2023) use diffusion models to overcome the challenges posed due to fusion inconsistency and semantic disharmony for efficient image composition. Avrahami et al. (2021) takes the target region mask and simply blends the noised version of the input image with local text-guided diffusion latent. Yang et al. (2023a) trains a diffusion model to blend an exemplar image on the source image at the position specified by an arbitrary shape mask and leverages the classifier-free guidance (Ho & Salimans, 2022) to increase the similarity to the exemplar image. Lu et al. (2023) introduces TF-ICON which leverages off-the-shelf diffusion models to perform cross-domain image-guided composition without

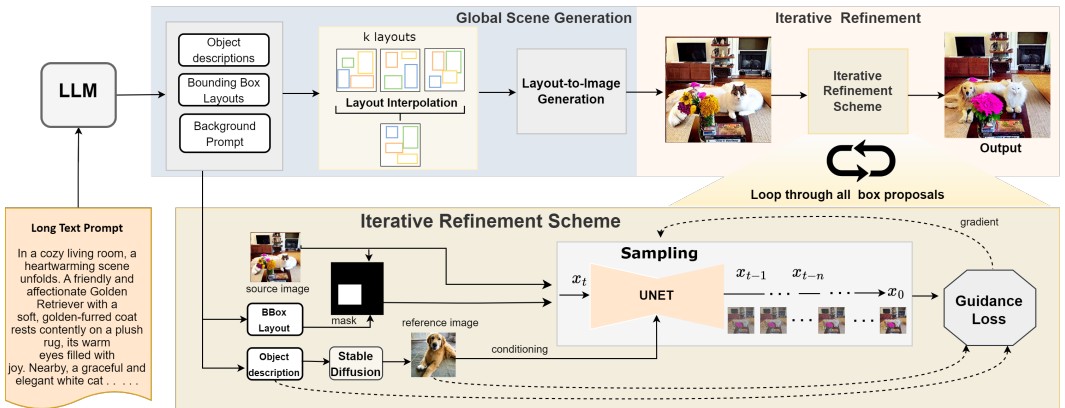

Figure 2: *Global Scene Generation:* Our proposed approach takes a long text prompt describing a complex scene and leverages an LLM to generate $k$ layouts which are then interpolated to a single layout, ensuring the spatial accuracy of object placement. Along with the layouts, we also query an LLM to generate object descriptions along with a concise background prompt summarizing the scene's essence. A Layout-to-Image model is employed which transforms the layout into an initial image. *Iterative Refinement Scheme:* The content of each box proposal is refined using a diffusion model conditioned on a box mask, a (generated) reference image for the box, and the source image, guided by a multi-modal signal.

requiring additional training, fine-tuning, or optimization. Based on these works, we build an iterative refining scheme, which performs region-based composition at the layout level, utilizing a given mask shape and modifies each layout based on the object characteristics guided by a multi-modal loss signal.

## 3 METHODOLOGY

### 3.1 PRELIMINARIES ON DIFFUSION MODELS.

Diffusion models are generative models that learn the data distribution of complex datasets. They consist of a forward diffusion process and a reverse diffusion process. During the forward process, noise is added to the input data point $x_0$ for $T$ steps, until the resulting vector $x_t$ is almost distributed according to a standard Gaussian distribution. Each step in the forward process is a Gaussian transition $q(x_t \mid x_{t-1}) := \mathcal{N}(\sqrt{1 - \beta_t} x_{t-1}, \beta_t \mathbf{I})$, where $\{\beta_t\}_{t=0}^{T}$ is a fixed or learned variance schedule. The resulting latent variable $x_t$ can be expressed as:

$$x_t = \sqrt{\alpha_t} x_0 + \sqrt{1 - \alpha_t} \epsilon, \quad \epsilon \sim \mathcal{N}(\mathbf{0}, \mathbf{I}), \tag{1}$$

where $\alpha_t := \prod_{s=1}^{t} (1 - \beta_s)$. The reverse process $q(x_{t-1} \mid x_t)$ is parametrized by another Gaussian transition $p_\theta(x_{t-1} \mid x_t) := \mathcal{N}(x_{t-1}; \mu_\theta(x_t, t), \sigma_\theta(x_t, t)\mathbf{I})$. $\mu_\theta(x_t, t)$ can be decomposed into the linear combination of $x_t$ and a noise approximation model $\epsilon_\theta(x_t, t)$, which is trained so that for any pair $(x_0, t)$ and any sample of $\epsilon$,

$$\epsilon_\theta(x_t, t) \approx \epsilon = \frac{x_t - \sqrt{\alpha_t} x_0}{\sqrt{1 - \alpha_t}}. \tag{2}$$

After training $\epsilon_\theta(x, t)$, different works (Song et al., 2020a; Song & Ermon, 2019; Song et al., 2020b) study different approximations of the unknown $q(x_{t-1} | x_t, x_0)$ to perform sampling. In our work, we utilize the denoising diffusion implicit model (DDIM) introduced by Song et al. (2020a) to predict the clean data point.

### 3.2 OVERVIEW

Our core objective is to generate images from long textual descriptions, ensuring that the resulting images faithfully represent the intricate details outlined in the input text. We generate the output image in a multi-step manner; generating an initial image that serves as a template at a global level, followed

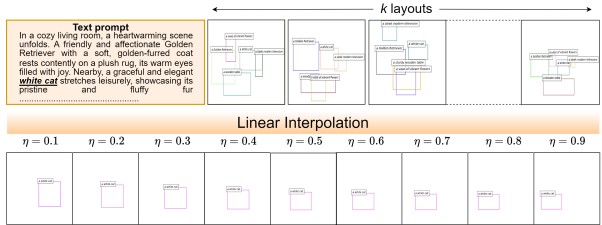

Figure 3: *Effect of interpolation factor $\eta$:* We interpolate the $k$ bounding boxes for each object and control the interpolation by the factor $\eta$. We visualize the change in the bounding box location of *"a white cat"* highlighted in the text for different $\eta$ values from 0.1 to 0.9 with increments of 0.1. Best viewed in zoom.

by a box-level refinement phase that serves as a corrective procedure. **1) Global scene generation:** We begin by decomposing lengthy text prompts into "Scene Blueprints." Scene Blueprints provide a structured representation of the scene containing: object bounding boxes in image space, detailed text descriptions for each box, and a background text prompt. We use an LLM to extract the Scene Blueprint from the given long prompt and use layout-conditioned text-to-image models to generate the initial image. Unlike prior works, we support additional user control on the box layouts by generating multiple proposal blueprints and providing an option to smoothly interpolate between the candidate layouts. This interpolation not only facilitates (optional) user control but also helps mitigate any potential errors introduced by the LLM. **2) Box-level content refinement:** In the second phase, we iterate through all the boxes and evaluate and refine their content in terms of quality and adherence to the prompt. We use a multi-modal guidance procedure that maximizes a quality score for each box.

## 3.3 GLOBAL SCENE GENERATION

Textual representation of a scene contains various characteristics that provide information about objects, their spatial properties, semantics, attributes, and more. To coherently capture these properties, we employ an off-the-shelf pre-trained large language model (LLM)(OpenAI, 2021). We instruct the LLM with an appropriately engineered prompt (see supplementary Sec. A.4) to generate a *Scene Blueprint* containing the following three components:

1. *Layout:* Bounding box coordinates for each object - $\{object : (x, y, width, height)\}$

2. *Object description:* Description associated with each object - $\{object : description\}$

3. *Background Prompt:* A general prompt describing the overall essence of the scene.

The layout and the background prompt generated by the LLM are then used to condition the diffusion model to generate an image. We follow recent work by Lian et al. (2023) which generates the image from the layouts in two steps, 1) generating masked latent inversion for each object bounding box, and 2) composing the latent inversion as well as generating the corresponding background from the background prompt.

**Layouts Interpolation and Noise Correction.** While LLMs have advanced spatial reasoning abilities, we observed that they struggle to model the spatial positions of the objects when presented with longer descriptions, often resulting in abnormal relative sizes of the objects and unnatural placement of object boxes in the scene (Sec.4.1). A naive approach to resolve these issues is to fine-tune an LLM on handcrafted data of text descriptions and bounding boxes. However, this approach requires extensive resources in terms of human annotators and compute requirements, and risks catastrophic forgetting (Luo et al., 2023). On the other hand, correcting these errors manually by adjusting the boxes is also a daunting task and defeats the purpose of using an LLM to extract layouts. To address these challenges, we propose a simple yet effective solution - *layout interpolation*. Instead of generating only one proposal layout, we query the LLM to generate $k$ layouts. Subsequently, we employ linear interpolation to compute the coordinates for each object's final bounding box, denoted as $o_j = (\hat{x}_j, \hat{y}_j, \hat{w}_j, \hat{h}_j)$, where for any coordinate $\hat{z}_j \in o_j$, $\hat{z}_j =\texttt{Interpolation}(z_j^{(1)}, z_j^{(2)}, z_j^{(3)} \ldots z_j^{(k)})$. The $\texttt{interpolation}$ function recursively updates each coordinate such that at any iteration $l$ of bounding boxes, $\hat{z}_j^{(l)} := (1 - \eta) \cdot \hat{z}_j^{(l-1)} + \eta \cdot z_j^{(l)}$, where $\eta$ is the interpolation factor which controls the influence of the individual bounding boxes on the final interpolated box (see Fig. 3).

As the long complex prompts may result in a large number of boxes, the images generated by the layout guidance tend to have color noise and small artifacts (See Fig. 8 in supplementary). Therefore, we optionally perform an image-to-image translation step using stable diffusion (Rombach et al., 2022), resulting in a cleaner image while preserving the semantics. We refer to this image as $x_{initial}$.

Despite conditioning on layout, we observe that the diffusion model is unable to generate all scene objects effectively. It struggles to compose multiple diverse objects having varied spatial and semantic properties in one shot. Consequently, $x_{initial}$ often has missing objects or fails to represent them accurately in accordance with their descriptions. Therefore, we employ a box-level refinement strategy that evaluates and refines the content within each bounding box of the layout in terms of quality and adherence to the prompt.

### 3.4 BOX-LEVEL REFINEMENT

Current diffusion models have certain limitations in terms of composing multiple objects in the image when presented with longer text prompts, a problem that is still unexplored. To overcome this issue and ensure faithful generation of all the objects, we introduce an *Iterative Refinement Scheme (IRS)*. Our proposed IRS works at the bounding box level and ensures the corresponding object at each bounding box is characterized by its properties given in the textual description. To achieve this, we iterate across each object's bounding box in $x_{initial}$ and compare the visual characteristics of the object with its corresponding description extracted from the text prompt. Consider an object $j$ with its bounding box $o_j = (\hat{x}_j, \hat{y}_j, \hat{w}_j, \hat{h}_j)$ and its textual characteristics denoted by $d_j$, we use CLIP score (Hessel et al., 2021) as a metric to get the similarity between the object and its description such that, $score(s) = \text{CLIP}(object(j), d_j)$. If the CLIP score $s$ is below a certain threshold, we modify the content of the bounding box such that it follows its corresponding description.

Any reasonable modification of the content within a bounding box must improve its fidelity and adherence to the prompt. Since diffusion models, e.g. stable diffusion (Rombach et al., 2022), are already good at generating high-fidelity images from shorter prompts, we exploit this ability and follow a paint-by-example approach. We generate a new object for the designated box by passing the object description $d_j$ to a Text-to-Image stable diffusion model. The generated image $x_j^{ref}$ acts as a reference content for the bounding box $o_j$. We then use a pretrained image composition model (Yang et al., 2023a) conditioned on the reference image $x_j^{ref}$, mask $m_j$ (extracted from the bounding box), and source image $x_{initial}$ to compose the reference object at the designated position on the source image specified by the mask. To ensure the composition generates the object that follows the properties described in the text prompt as closely as possible, we guide the sampling process by an external multi-modal loss signal.

**Box-level Multi-modal Guidance.** Given the initial image $x_{initial}$, a reference image $x_j^{ref}$, a guiding text prompt $d_j$ and a binary mask $m_j$ that marks the region of interest in the image corresponding to bounding box $o_j$, our goal is to generate a modified image $\hat{x}$, s.t. the content of the region $\hat{x} \odot m_j$ is consistent with the prototype image $x_j^{ref}$ and adheres to the text description $d_j$, while the complementary area remains as close as possible to the source image, i.e., $x_{initial} \odot (1 - m_j) \approx \hat{x} \odot (1 - m_j)$, where $\odot$ is the element-wise multiplication. Dhariwal & Nichol (2021) use a classifier trained on noisy images to guide generation towards a target class. In our case, we use an external function in the form of CLIP to guide the generation in order to adhere to the prototype image $x_j^{ref}$ and text description $d_j$. However, CLIP is trained on noise-free data samples, we estimate a clean image $\boldsymbol{x_0}$ from each noisy latent $\boldsymbol{x_t}$ during the denoising diffusion process via Eqn. 1 as follows,

$$\hat{\boldsymbol{x_0}} = \frac{\boldsymbol{x_t} - (\sqrt{1 - \alpha_t})\boldsymbol{\epsilon_\theta}(\boldsymbol{x_t}, t)}{\sqrt{\alpha_t}}. \tag{3}$$

Our CLIP based multimodal guidance loss can then be expressed as,

$$\mathcal{L}_{CLIP} = \mathcal{L}_{cosine}\Big(\text{CLIP}_{\text{image}}(\hat{x}_0 \odot m_j), \text{CLIP}_{\text{text}}(d_j)\Big) + \\ \lambda \cdot \mathcal{L}_{cosine}\Big(\text{CLIP}_{\text{image}}(\hat{x}_0 \odot m_j), \text{CLIP}_{\text{image}}(x_j^{ref})\Big), \tag{4}$$

where $\mathcal{L}_{cosine}$ denotes the cosine similarity loss and $\lambda$ is a hyperparameter. The first part of the equation measures the cosine loss between the composed object at the region specified by the mask

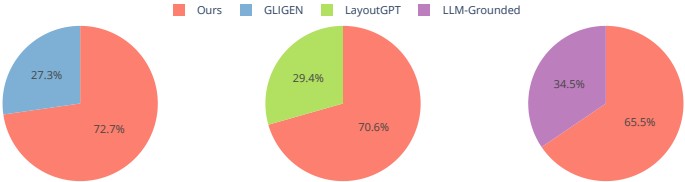

Figure 4: User study. A majority of users picked our method compared to prior works when presented with a 2-AFC task of selecting the image that adheres to the given prompt the most.

$m_j$ and its corresponding text characteristics $d_j$. The second part of the equation measures the cosine loss between the composed object and its corresponding prototype $x_j^{ref}$. A similar approach using CLIP text-based guidance is used in Avrahami et al. (2021) for region-based modification. However, in contrast, we also include a CLIP image-based guidance to steer the generation toward the prototype image $x_j^{ref}$ to account for the fine-grained details that may not be captured in the text description. In order to confine the modification within the given bounding box, we optionally employ a background preservation loss $\mathcal{L}_{bg}$ which is a summation of the L2 norm of the pixel-wise differences and Learned Perceptual Image Patch Similarity metric Zhang et al. (2018b) between $\widehat{x}_0 \odot (1 - m_j)$ and $x_{initial} \odot (1 - m_j)$ The final diffusion guidance loss is thus the weighted sum of $\mathcal{L}_{CLIP}$ and $\mathcal{L}_{bg}$ given as,

$$\mathcal{L}_{guidance} = \mathcal{L}_{CLIP} + \gamma \cdot \mathcal{L}_{bg}. \quad (5)$$

The gradient of the resultant loss $\nabla_{\widehat{x}_0} \mathcal{L}_{guidance}$ is used to steer the sampling process to produce an object at the bounding box $o_j$ which follows the properties of prototype $x_j^{ref}$ and description $d_j$. Additionally, refer to the supplementray section A.8 to understand how guidance loss influences the generation process and algorithm A.1 for a full end-to-end pipeline.

## 4 EXPERIMENTS

**Settings:** Our framework uses a combination of several components. For acquiring the long text descriptions, we ask ChatGPT to generate scenes on various themes. In addition to this, we also use the textual descriptions from some COCO (Lin et al., 2014) and PASCAL (Everingham et al., 2010) images by querying an image captioning model (Zhu et al., 2023) to generate a detailed description spanning 80-100 words. For extracting layouts, bounding boxes, and background prompt, we make use of ChatGPT completion api (OpenAI, 2021) with an appropriate instruction template (See Supplementary Sec. A.4). We generate 3 layouts for each text prompt and interpolate them to a single layout to account for the spatial location correctness. To avoid layout overlap, we push the boxes away from each other until there is minimal contact wherever feasible. For base layout-to-image generation, we use the work of Lian et al. (2023) and scale it for longer text prompts. We use 20 diffusion steps at this point. For box refinement, we use the pre-trained image composition model of Yang et al. (2023a) which conditions on a reference image. For each box refinement, we use 50 diffusion steps. For implementation, we use Pytorch 2.0. Finally, our entire pipeline runs on a single Nvidia A100 40GB GPU.

**Quantitative Results:** Our work stands as a first effort to address the challenge of generating images from extensive text prompts. As discussed in Section 1, current diffusion-based text-to-image generation methods typically utilize the CLIP tokenizer to condition the diffusion model for image generation. However, this approach can lead to inconsistent images when confronted with lengthy text prompts with intricate details. To the best of our knowledge, there is currently no established metric for assessing the performance of diffusion models in handling lengthy text descriptions. Hence we propose to use the *Prompt Adherence Recall (PAR) score* to quantify adherence to the prompt defined as *Mean of **object presence** over all objects over all prompts* where we use an off-the-shelf object detector (Li et al., 2022b) to check if the object is actually present in the generated image such that *object presence* is 1 if present and 0 otherwise. We achieve a PAR score of $85\%$ which is significantly better than Stable Diffusion ($49\%$), GLIGEN ($57\%$) and LayoutGPT ($69\%$). We also conducted an extensive user study to assess the effectiveness of our method in comparison to four established baseline approaches: GLIGEN (Li et al., 2023), LayoutGPT (Feng et al., 2023) and LLM-Grounded Diffusion (Lian et al., 2023). To mitigate any bias, participants were instructed to

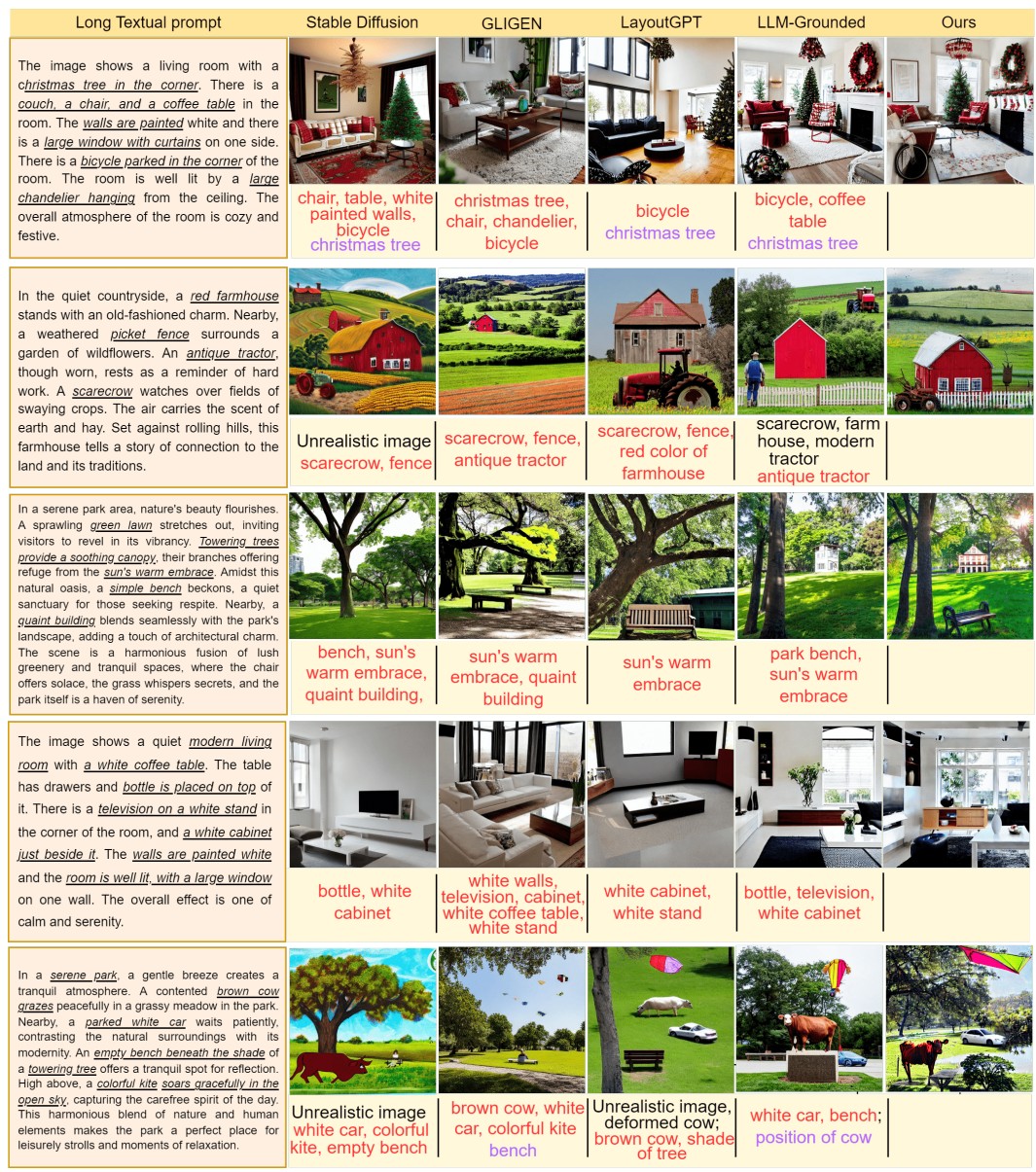

Figure 5: *Qualitative comparisons:* We compare our image generation method to state-of-the-art baselines, including those using layouts. The underlined text in the text prompts represents the objects, their characteristics, and spatial properties. Red text highlights missing objects, purple signifies inaccuracies in object positioning, and black text points out implausible or deformed elements. Baseline methods often omit objects and struggle with spatial accuracy (first four columns), while our approach excels in capturing all objects and preserving spatial attributes (last column).

select one image from a pair of images randomly selected from two distinct approaches. Their goal was to choose the image that most accurately represented the provided textual descriptions regarding spatial arrangement, object characteristics, and overall scene dynamics. The outcomes of the user study are presented in Fig. 4. Our findings demonstrate that, on average, our proposed method consistently produces coherent images that closely align with their respective textual descriptions, whereas existing approaches struggle to effectively handle longer text prompts. For a detailed procedure and user-study results on the fidelity, refer to section A.9 of supplementary.

**Qualitative Analysis:** Fig. 5 presents a qualitative assessment of our method in comparison to established state-of-the-art methods. The text descriptions include specific phrases denoted by underlined italics, conveying information about objects, their attributes, and spatial arrangements.

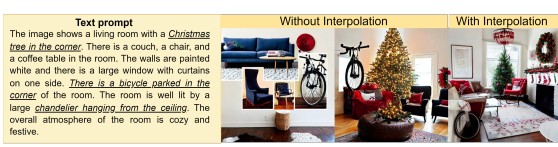

Figure 6: *Effect of Layout Interpolation:* Our layout interpolation method (last column) significantly improves object spatial positioning compared to non-interpolated cases (first two columns). Best viewed in zoom.

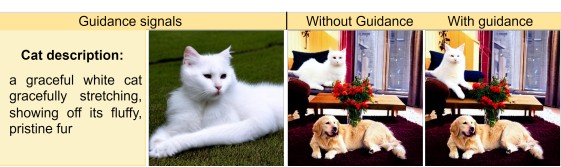

Figure 7: *Effect of Guidance:* Without guidance signal, the composed image does not follow the properties corresponding to its description and visual appearance. In contrast, the one with the guidance (right) adheres to the visual prototype and description.

Notably, the red text beneath each image highlights instances of missing objects, while the purple text indicates spatial inaccuracies, and the black text identifies elements of implausibility or distortion. From the figure, the stable diffusion baseline (column 1) frequently falls short in incorporating certain objects from the prompt due to its inability to efficiently handle lengthy text prompts. In some instances (rows 2 and 4), the generated images exhibit unrealistic features. Layout-based approaches (columns 2, 3, and 4) also encounter difficulties in fully capturing the nuances of the text prompt, resulting in occasional omissions of objects and instances of deformations (column 3, row 5). In contrast, our approach excels by accurately capturing all objects from the text, including their intricate details and precise spatial positions. Refer to sections A.3 and A.5 for further results and comparisons with additional baselines.

## 4.1 ABLATIONS

**Effect of Layout Interpolation.** We query the LLM to generate multiple layouts from the input text prompt and then employ linear interpolation to merge them into a single layout. However, for lengthy prompts, LLMs can occasionally generate layouts with random object placements, resulting in unnatural images. As shown in Fig. 6, the first two columns depict images without layout interpolation, while the last column shows the interpolated image. The underlined phrases in the text prompt indicate object spatial characteristics. In contrast, the last column demonstrates the improved result with interpolation, aligning nearly every object with its textual spatial description.

**Effect of Guidance** The external guidance in the form of CLIP multi-modal loss used in the refinement stage steers sampling of the specific box proposal towards its corresponding description and reference prototype. We present a visual illustration of this phenomenon in Fig. 7. As seen from the figure, the properties of the cat get more aligned with the prototype image and text description in the presence of a guidance signal.

## 5 CONCLUSION

In this work, we identified the limitations of prior text-to-image models in handling complex and lengthy text prompts. In response, we introduced a framework involving a data structure (*Scene Blueprint*) and a multi-step procedure involving *global scene generation* followed by an *iterative refinement scheme* to generate images that faithfully adhere to the details in such lengthy prompts. Our framework offers a promising solution for accurate and diverse image synthesis from complex text inputs, bridging a critical gap in text-to-image synthesis capabilities.

While we presented a simple interpolation technique to combine various bounding box proposals, we maintained fixed box layouts for the second phase. A promising avenue for future research lies in exploring dynamic adjustments of boxes within the iterative refinement loop. Another area warranting further examination pertains to the handling of overlapping boxes. While we currently address this challenge by sorting boxes by size prior to the box-level refinement phase, there is an opportunity to explore more advanced techniques for managing overlaps. Additionally, our current approach to box-level refinement treats each object in isolation, overlooking the relationships that exist among objects within a scene. A compelling avenue for future research is to incorporate and leverage these object relationships, with the aim of achieving more comprehensive and contextually aware image generation.

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

## A APPENDIX

### A.1 ALGORITHM

In Sec. 3, we describe our proposed approach *LLM Blueprint*. Our method uses an LLM to first generate bounding box layouts and descriptions pertaining to each object in the description, along with a background prompt that describes the overall scene in the text. We use specific instruct prompts to do so (see Appendix.A.4). Due to possibly inconsistent box positions, we query the LLM to generate $k$ layouts giving $k$ bounding boxes for each object in the layout. We then linearly interpolate the coordinates to obtain one final bounding box layout denoted as $\mathcal{O}$. We use a layout-to-image generation model $\mathcal{M}_L$ to obtain an initial image. However, as described in Sec.3.3 of the main paper, we apply an additional noise correction step by passing the generated image to an image-to-image diffusion model $SD_{image-image}$ while retaining the semantics of the scene. We call the image generated until this point as *Global Scene Generation* and denote it as $x_{initial}$. However, the image generated after the first phase $x_{initial}$ still has inconsistencies in generating objects as per their characteristics. To overcome this issue and ensure faithful generation of all the objects, we introduce an *Iterative Refinement Scheme (IRS)*. Our proposed method works at the bounding box level and ensures the corresponding object at each bounding box is characterized by its properties given in the textual description. We achieve this by looping across each bounding box and iteratively modifying the regions based on a CLIP metric. In other words, we generate a new object for the designated box by passing the object description $d_j$ to a text-to-image stable diffusion model $SD_{text-image}$. The generated image $x_j^{ref}$ acts as a reference content for the bounding box $o_j$. We further create a binary mask $m_j$ from the bounding box coordinates. We then use an image composition model $\mathcal{M}_{comp}$ $(\mu_\theta(x_t), \Sigma_\theta(x_t))$ conditioned on the reference image $x_j^{ref}$, mask $m_j$ and source image $x_{initial}$ to compose the reference object at the designated position on the source image specified by the mask $m_j$. To ensure the composition generates the object that exactly follows the properties described in the text prompt, we use an external function in the form of CLIP to guide the generation in order to adhere to the prototype image $x_j^{ref}$ and text description $d_j$. However, since our prototype $x_j^{ref}$ is in the input space and CLIP is trained on real-world data samples, we estimate a clean image $x_0$ from each noisy latent $x_t$ during the denoising diffusion process via Eqn. 1 (main paper). We optionally add a background preservation loss $\mathcal{L}_{bg}$ to avoid the composition process affecting the background. We provide a pseudo code of our algorithm in Algorithm 1.

### A.2 NOISE AND ARTIFACT CORRECTION

As discussed in Sec. 3.3, the generated image after the first phase sometimes suffers from noise or artifacts. We do an optional noise correction step after Global scene generation, which essentially removes the unwanted noise and artifacts. Specifically, we utilize the image-to-image translation method of stable diffusion Rombach et al. (2022) which instead of starting from random noise, starts from an input image, adds noise to it and then denoises it in the reverse process. The idea is to enhance the quality of the image while maintaining its semantics. We notice that this process removes the unwanted noise and artifacts present in the image (See Figure 8).

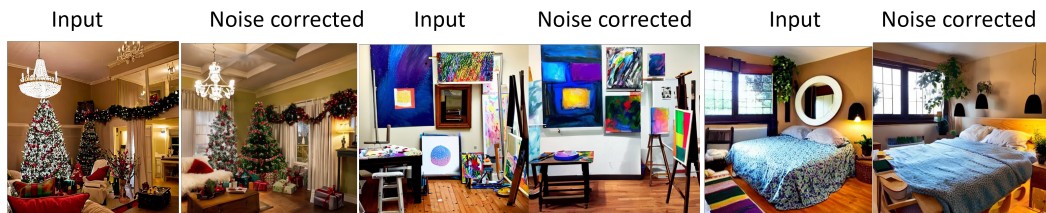

Figure 8: *Noise Correction:* The noise correction strategy removes noise and certain redundant artifacts from the image that are unnecessary.

### A.3 FURTHER QUALITATIVE COMPARISONS

We provide further qualitative comparisons of our approach with state-of-the-art baselines in Fig. 9.

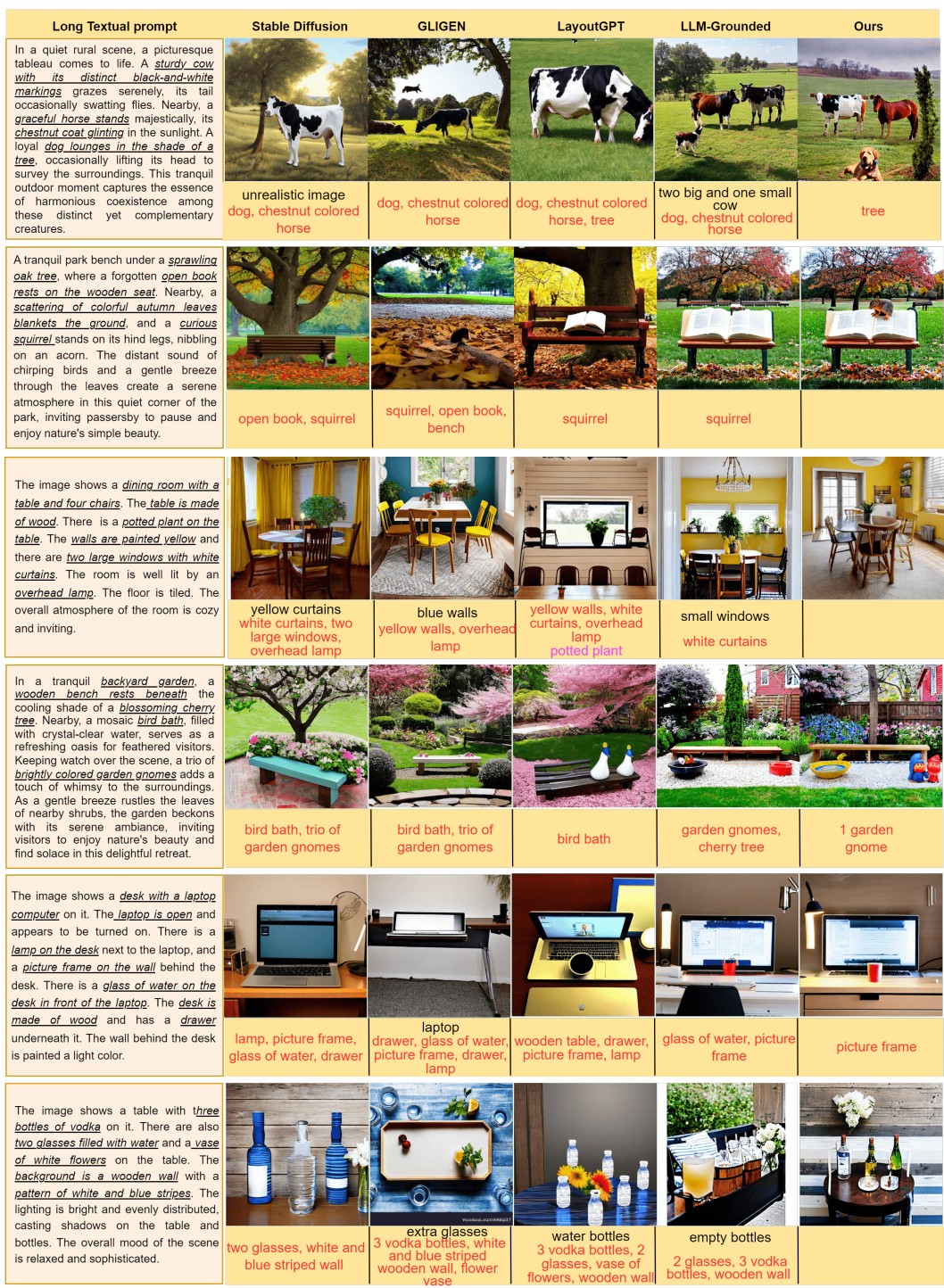

Figure 9: *Qualitative comparisons:* We provide further qualitative comparisons to our approach against the state-of-the-art baselines. The underlined text in the text prompts represents the objects, their characteristics, and spatial properties. Baseline methods often omit objects and struggle with spatial accuracy (first four columns), while our approach excels in capturing all objects and preserving spatial attributes (last column).

---

**Algorithm 1** LLM Blueprint

---

**Input:** Long textual description $\mathcal{C}$, diffusion steps $k$, sampling iterations $n$, layout-to-image model $\mathcal{M}_L$, $LLM$, image composition model $\mathcal{M}_{comp}$ $(\mu_\theta(x_t), \Sigma_\theta(x_t))$, LLM queries $q$, stable diffusion model $SD$, $CLIP$ model, **Interpolation** function.

**Output:** Composed image $\widehat{x}$ encompassing all the elements of the textual description $\mathcal{C}$.

// Get layouts, object details, and background prompt from LLM.

$\mathcal{O}|_k, \mathcal{D}, \mathcal{P}_b \leftarrow LLM(\mathcal{C})$

// Interpolate layouts.

$\mathcal{O} \leftarrow \textbf{Interpolation}(\mathcal{O}|_k)$

// Generate initial image from interpolated layout.

$x_{\text{initial}} \leftarrow \mathcal{M}_L(\mathcal{O}, \mathcal{D}, \mathcal{P}_b)$

// Optional noise correction.

$x_{\text{initial}} \leftarrow SD_{\text{image-image}}(x_{\text{initial}})$

// Iterative Refinement.

**foreach** $(o_j, d_j)$ *in* $\{\mathcal{O}, \mathcal{D}\}$ **do**

$\quad x_j^{ref} \leftarrow SD_{\text{text-image}}(d_j)$

$\quad m_j \leftarrow GetMask(o_j)$

$\quad x_k \sim \mathcal{N}(\sqrt{\bar{\alpha}_k}x_0, (1 - \bar{\alpha}_k)\mathbf{I})$

$\quad$ **for** $t$ *from* $T$ *to* 1 **do**

$\quad\quad$ **for** *iter from 1 to n* **do**

$\quad\quad\quad \mu, \Sigma \leftarrow \mu_\theta(x_t), \Sigma_\theta(x_t)$

$\quad\quad\quad \widehat{x}_0 \leftarrow \frac{x_t}{\sqrt{\bar{\alpha}_t}} - \frac{\sqrt{1 - \bar{\alpha}_t}\epsilon_\theta(x_t, t)}{\sqrt{\bar{\alpha}_t}}$

$\quad\quad\quad \mathcal{L} \leftarrow \mathcal{L}_{CLIP}(\widehat{x}_0, d_j, m_j) + \lambda\mathcal{L}_{CLIP}(\widehat{x}_0, x_j^{ref}, m_j) + \gamma\mathcal{L}_{bg}(x, \widehat{x}_0, m_j)$

$\quad\quad\quad \hat{\epsilon} \leftarrow \epsilon_\theta(x_t) - \sqrt{1 - \bar{\alpha}_t}\nabla_{\widehat{x}_0}\mathcal{L}$

$\quad\quad\quad x_{\text{fg}} \leftarrow \sqrt{\bar{\alpha}_{t-1}}\left(\frac{x_t - \sqrt{1 - \bar{\alpha}_t}\hat{\epsilon}}{\sqrt{\bar{\alpha}_t}}\right) + \sqrt{1 - \bar{\alpha}_{t-1}}\hat{\epsilon}$

$\quad\quad\quad x_{t-1} \leftarrow m_j \odot x_{\text{fg}} + (1 - m_j) \odot \mathcal{N}(\sqrt{\bar{\alpha}_t}x_0, (1 - \bar{\alpha}_t)\mathbf{I})$

$\quad x_0 \leftarrow x_{t-1}$

**return** $x_0$

---

## A.4 INSTRUCT PROMPTS FOR LLM

Our approach utilizes LLMs' advanced spatial and reasoning capabilities to derive layouts, object descriptions, and background prompt from a long textual description. For extracting layouts and background prompts, we use the prompt designed by Lian et al. (2023) and scale it to work on longer textual descriptions. For extracting object descriptions, we designed our own unique prompt. Below are the instruct prompts utilized in our work.

**Instruct prompt for extracting layouts and background prompt.**

---

You are an intelligent bounding box generator. I will provide you with a caption for a photo, image, a detailed scene, or a painting. Your task is to generate the bounding boxes for the objects mentioned in the caption, along with a background prompt describing the scene. The images are of size 512x512. The top-left corner has coordinates [0, 0]. The bottom-right corner has coordinates [512, 512]. The bounding boxes should not overlap or go beyond the image boundaries. Each bounding box should be in the format of (object name, [top-left x coordinate, top-left y coordinate, box width, box height]) and include exactly one object (i.e., start the object name with "a" or "an" if possible). Do not put objects that are already provided in the bounding boxes into the background prompt. Do not include non-existing or excluded objects in the background prompt. If needed, you can make reasonable guesses. Please refer to the example below for the desired format.

Caption: In the quiet countryside, a red farmhouse stands with an old-fashioned charm. Nearby, a weathered picket fence surrounds a garden of wildflowers. An antique tractor, though worn, rests as a reminder of hard work. A scarecrow watches over fields of swaying crops. The air carries the scent of earth and hay. Set against rolling hills, this

---

farmhouse tells a story of connection to the land and its traditions
Objects: [('a red farmhouse', [105, 228, 302, 245]), ('a weathered picket fence', [4, 385, 504, 112]), ('an antique tractor', [28, 382, 157, 72]), ('a scarecrow', [368, 271, 66, 156]) ]
Background prompt: A realistic image of a quiet countryside with rolling hills

Caption: A realistic image of landscape scene depicting a green car parking on the left of a blue truck, with a red air balloon and a bird in the sky
Objects: [('a green car', [21, 181, 211, 159]), ('a blue truck', [269, 181, 209, 160]), ('a red air balloon', [66, 8, 145, 135]), ('a bird', [296, 42, 143, 100])]
Background prompt: A realistic image of a landscape scene

---

**Instruct prompt for extracting object descriptions.**

---

You are an intelligent description extractor. I will give you a list of the objects and a corresponding text prompt. For each object, extract its respective description or details mentioned in the text prompt. The description should strictly contain fine details about the object and must not contain information regarding location or abstract details about the object. The description must also contain the name of the object being described. For objects that do not have concrete descriptions mentioned, return the object itself in that case. The output should be a Python dictionary with the key as object and the value as description. The description should start with 'A realistic photo of object' followed by its characteristics. Sort the entries as per objects that are spatially behind (background) followed by objects that are spatially ahead (foreground). For instance object "a garden view" should precede the "table". Make an intelligent guess if possible. Here are some examples:

list of objects: [a Golden Retriever,a white cat,a wooden table,a vase of vibrant flowers,a sleek modern television]
text prompt: In a cozy living room, a heartwarming scene unfolds. A friendly and affectionate Golden Retriever with a soft, golden-furred coat rests contently on a plush rug, its warm eyes filled with joy. Nearby, a graceful and elegant white cat stretches leisurely, showcasing its pristine and fluffy fur. A sturdy wooden table with polished edges stands gracefully in the center, adorned with a vase of vibrant flowers adding a touch of freshness. On the wall, a sleek modern television stands ready to provide entertainment. The ambiance is warm, inviting and filled with a sense of companionship and relaxation.

output: {a sleek modern television: A realistic photo of a sleek modern television., a wooden table: A realistic photo of a sturdy wooden table with polished edges., vase of vibrant flowers: A realistic photo of a vase of vibrant flowers adding a touch of freshness., a Golden Retriever: 'A realistic photo of a friendly and affectionate Golden Retriever with a soft, golden-furred coat and its warm eyes filled with joy., a white cat: 'A realistic photo of a graceful and elegant white cat stretches leisurely, showcasing its pristine and fluffy fur.}

## A.5 ADDITIONAL COMPARISONS WITH DEEPFLOYD AND DENSEDIFFUSION

As recommended we provide visual comparison of our method with DeepFloyd StabilityAI (2023) and DenseDiffusion Kim et al. (2023) in Fig. 10. As seen from the figure, DeepFloyd with a strong T5 text encoder struggles to generate coherent images with complex compositional prompts. The same is true for DenseDiffusion. We conclude that our scene blueprint augmented with iterative refinement is necessary for generating coherent images from complex compositional prompts.

We further present quantitative comparison in terms of Prompt Adherence Recall (PAR) (see Sec. 4 of main paper) of our approach with all the baselines in Table 1. We also report average inference time for each approach. As seen from the table, DeepFloyd with a PAR score of 60% is highly inefficient as it takes around 8 min to generate a 256x256 image from a long textual prompt. We notice that while other approaches are slightly efficient in time, they report a lower PAR score, thus rendering them ineffective on complex compositional prompts. Our approach with an average inference time of around 3 min (including blue print generation and iterative refinement process) has the highest PAR

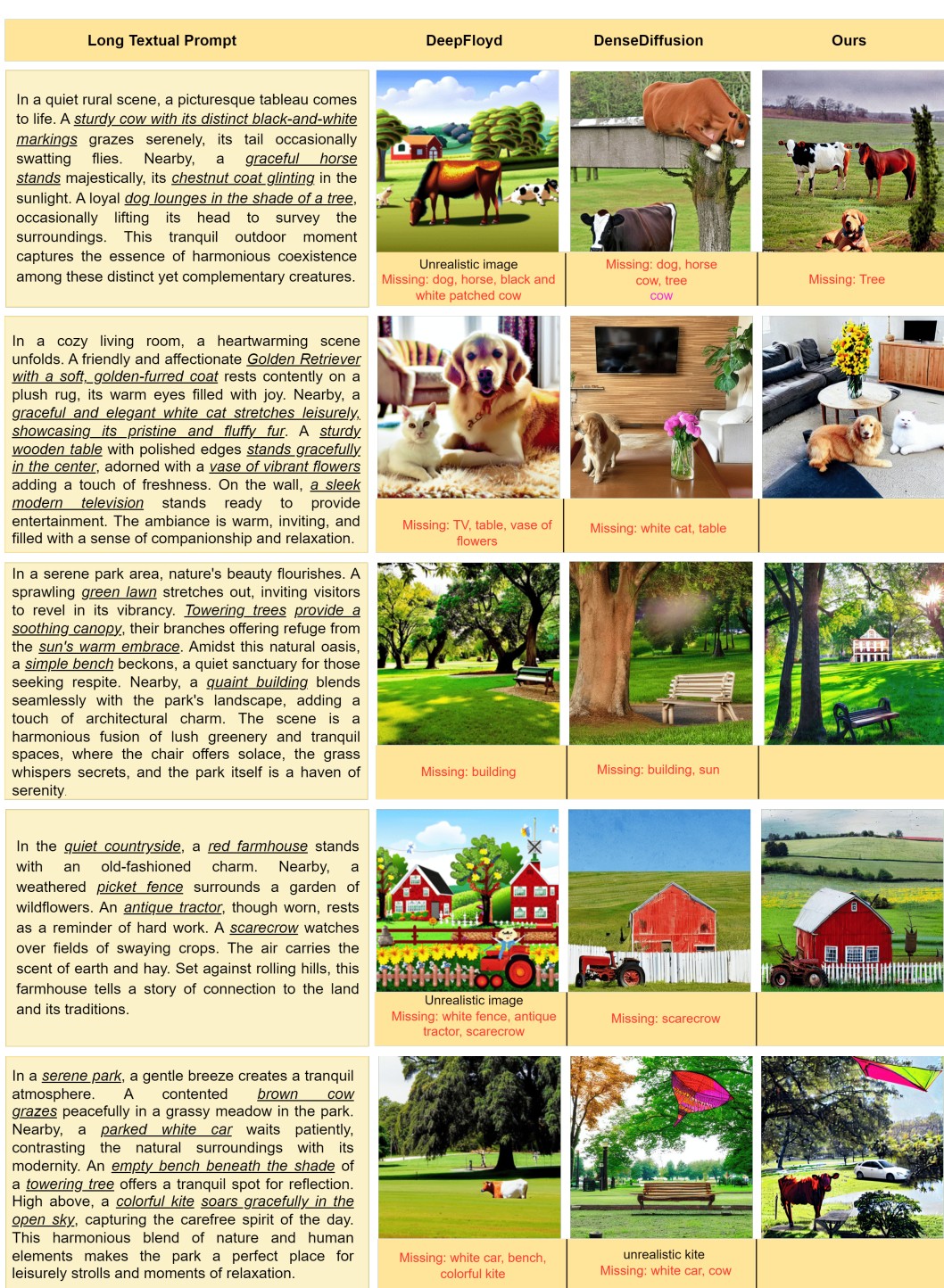

Figure 10: We provide qualitative comparisons of our approach against DeepFloyd and DenseDiffusion. The underlined text in the text prompts represents the objects, their characteristics, and spatial properties. Baseline methods often omit objects and struggle with spatial accuracy (first 2 columns), while our approach (last column) excels in capturing all objects and preserving spatial attributes. The text in black below images (if present) shows the unrealistic nature of image, red text enlists the missing objects in the image and pink text refers to location misalignment of the object.

Table 1: **Quantitative comparison of Our approach with baselines in terms of PAR score and average inference time.** Our approach has the best PAR score while having relatively decent efficiency in terms of inference time.

| Method | PAR score (%) ↑ | Inference time (min.) ↓ |
|---|---|---|
| Stable Diffusion | 49 | 0.18 |
| GLIGEN | 57 | 0.50 |
| LayoutGPT | 69 | 0.83 |
| DenseDiffusion | 52 | 2.50 |
| DeepFloyd | 60 | 8.33 |
| Ours | **85** | **3.16** |

**Text prompt**

In a cozy living room, a heartwarming scene unfolds. A friendly and affectionate Golden Retriever with a soft, golden-furred coat rests contently on a plush rug, its warm eyes filled with joy. Nearby, on the right a graceful and elegant white cat stretches leisurely, showcasing its pristine and fluffy fur. A sturdy wooden table with polished edges stands gracefully in the center, adorned with a vase of vibrant flowers adding a touch of freshness. On the wall, a sleek modern television stands ready to provide entertainment. The ambiance is warm, inviting, and filled with a sense of companionship and relaxation.

| K=1 | K=2 | K=3 | K=4 | K=5 |
|---|---|---|---|---|

Figure 11: **Effect of number of layouts on final generated image.** We notice that final generated image is coherent in all the cases and aligns well with the textual prompt in terms of object attributes and spatial positions. Note that the position of cat and dog are well defined from the text i.e. cat is on the right of dog.

score of 85%, thus validating the effectiveness of our approach. Therefore, a discernible trade-off emerges between addressing complexity and ensuring the faithful reproduction of images.

## A.6 EFFECT OF NUMBER OF LAYOUTS ON FINAL GENERATED IMAGE

We present an analysis to study the effect of number of layouts on the final generated image in Fig 11. Consistent with the findings of Li et al. (2022b), the layouts generated by LLM (ChatGPT) most of the times align with the textual prompts. The interpolation of multiple layouts (K>1) produces coherent images preserving the spatial relationships between the objects i.e. dog is always towards

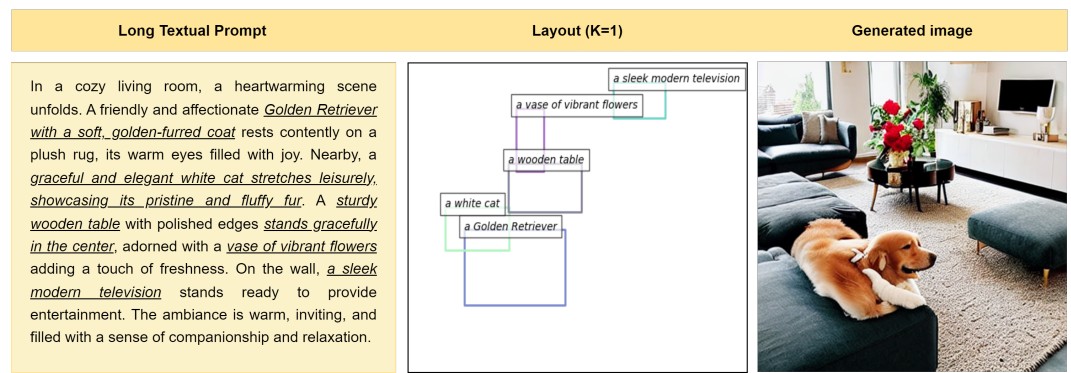

Figure 12: **Effect of Interpolation:** While ChatGPT is good at generating layouts from textual prompts, however, in few cases it can generate misaligned layouts (2nd row), leading to images missing certain objects such as cat (last row). We notice that the position of cat and dog are not well defined from the textual prompt.

Table 2: **Effectiveness of ChatGPT in modeling spatial relationships.** We observe that ChatGPT perfectly follows the object positions given prompt with clearly defined object positions. While as it shows inherent bias for the prompts with ambiguous object poistions.

| Prompt | Object | right | left | arbitrary position |
|---|---|---|---|---|
| A living room with a cat and a dog sitting one on each side | dog | 0.3 | 0.6 | 0.1 |
| | cat | 0.7 | 0.2 | 0.1 |
| A living room with a cat sitting towards right and a dog sitting towards left | dog | 0 | 1 | 0 |
| | cat | 1 | 0 | 0 |
| A living room with a dog sitting towards right and a cat sitting towards left | dog | 1 | 0 | 0 |
| | cat | 0 | 1 | 0 |

the left while cat is on the right. However, in extreme cases with only one layout, such as in Fig 12, the ChatGPT can sometimes generate spatially incorrect boxes, such as that for dog and cat, leading to missing objects or incorrect spatial positions of the objects in the final generated image (Fig. 6 of main paper)

## A.7 EFFECTIVENESS OF CHATGPT IN MODELING SPATIAL RELATIONSHIPS

Consistent with the findings of Li et al. (2022b), we observed that proprietary LLMs such as ChatGPT are exceptionally good at following the object positions from the textual prompt. We conducted an analysis in Table 2 with ChatGPT to verify its effectiveness on well-defined and ambiguous prompts (position of some objects is unclear). Specifically we prompted ChatGPT to generate bounding boxes of cat and dog with three different prompts: "A living room with a cat and a dog sitting on each side", "A living room with a cat sitting towards right and a dog sitting towards left", "A living room with a dog sitting towards right and a cat sitting towards left". Our analysis reveals that on an average 60% of the times for the ambiguous prompt "A living room with a cat and a dog sitting on each side", the chatGPT generates bounding box on the right for the cat and on left for the dog. While for other two unambiguous prompts, the ChatGPT generates correct location of bounding boxes for cat and dog. This shows that ChatGPT works exceptionally well for unambiguous prompts with clearly defined spatial relationships. For the ambiguous prompt, we notice an inherent bias inside ChatGPT, which leads to it generating dog on the left and cat on the right. To account for the errors and to provide a meaningful fix to the inherent bias inside ChatGPT, we provide a hyperparameter $\eta$ in the interpolation, which can be controlled to adjust for the bounding box of each object (see Fig. 3 in the main paper for a visual illustration of effect of $\eta$ parameter).

## A.8 SAMPLING PROCESS WITH GUIDANCE

Kindly note that the guidance function enables diffusion models to be controlled by arbitrary guidance modalities without the need to retrain any specific components (Bansal et al., 2023). Since ours is a training-free approach, therefore, to ensure the objects at each bounding box location follow their corresponding description in the long textual prompt, we employ CLIP-based loss $\mathcal{L}_{CLIP}$ at each step $t$ during sampling process (Eq. 5, main paper). During sampling, we first compute the gradient of this loss with respect to the estimated clean sample $\widehat{x}_0$ from Eq. 3 of the main paper and denote it as $\nabla_{\widehat{x}_0}\mathcal{L}_{CLIP}$. Using this gradient, we update the noise $\epsilon$ such that the updated noise at time step $t$ is expressed as,

$$\hat{\epsilon} \leftarrow \epsilon_\theta(x_t) - \sqrt{1 - \bar{\alpha}_t}\nabla_{\widehat{x}_0}\mathcal{L}_{CLIP} \tag{6}$$

Based on the above-updated noise, the sampling step is given as,

$$x_{fg} \leftarrow \sqrt{\bar{\alpha}_{t-1}}\left(\frac{x_t - \sqrt{1 - \bar{\alpha}_t}\hat{\epsilon}}{\sqrt{\bar{\alpha}_t}}\right) + \sqrt{1 - \bar{\alpha}_{t-1}}\hat{\epsilon}, \tag{7}$$

where $x_{fg}$ represents the latent variable of the image containing the specific foreground object being refined. This is combined with the unaltered background to get the final output latent as shown below,

$$x_{t-1} \leftarrow m \odot x_{fg} + (1 - m) \odot \mathcal{N}(\sqrt{\bar{\alpha}_t}x_0, (1 - \bar{\alpha}_t)\mathbf{I}), \tag{8}$$

where $m$ represents the mask for the object being refined. The process is repeated $n$ iterations across $T$ time steps and finally decoded to get the image with refined object. Further please refer to the Algorithm A1 in Appendix A1 for a full end-to-end pipeline.

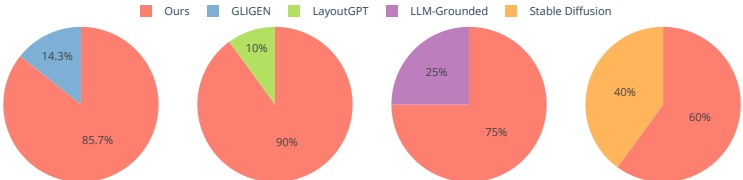

Figure 13: **User study on quality.** A majority of users picked our method compared to prior works when presented with a 2-AFC task of selecting the image with highest fidelity.

## A.9   STUDY ON FIDELITY

**Details of user study**. The subjects for human study were recruited from a pool of academic students with backgrounds in computer science and engineering. To ensure an unbiased evaluation and maintain objectivity, the subjects participating in the evaluation remained anonymous with no conflicts of interest with the authors. Furthermore, we prioritized user confidentiality by refraining from collecting personal details such as names, addresses etc. The entire survey process was anonymized, ensuring that there was no leakage of data to any user. Following a 2-AFC (two-alternative forced choice) design, each participant was tasked with selecting one image out of a pair of images, having highest fidelity. Each pair of images was randomly sampled from a pool containing images from baselines as well as our approach. This system mitigates biases which can originate from user-curated surveys. Further, the pool of images was shuffled before sampling each time. The system was further equipped with the feature of randomly flipping the position of the images, to remove the user-specific position bias in case any pair of images is repeated. Additionally, a 2-second delay between questions was introduced to facilitate more thoughtful decision-making.

Following above procedure, we present an extensive user study in Fig. 13 on the fidelity of generated images. For the user study, we compared our method with Stable Diffusion, GLIGEN, LayoutGPT and LLM Grounded Diffusion. The system yielded approximately 90 responses from 15 subjects, with each user answering an average of 6 questions. From the Fig. 13, it's clear that our approach compares favorably against baselines in terms of image fidelity while maintaining highest alignment with the text.

