# OpenReview forum: "LLM Blueprint: Enabling Text-to-Image Generation with Complex and Detailed Prompts"
_ICLR.cc/2024/Conference — ICLR 2024 poster_

### Official Review · Reviewer_kt1C · 2023-10-27

**Soundness:** 3 good
**Presentation:** 3 good
**Contribution:** 2 fair
**Rating:** 6
**Confidence:** 4

**Summary:**

This proposed a framework for generating images from complex and detailed prompts, which took more work for previous models or frameworks. They utilize LLM's capability to generate augmented textual and visual prompts for better generations. And the framework comprises global scene generation and an iterative refinement scheme to align with conditional cues.

**Strengths:**

- This work steps toward longer textual descriptions for image generations to ensure the fidelity of complex textual prompts.

- Scene blueprints with the iterative refinement step ensure high prompt adherence recall, quantitatively validating its effectiveness.

**Weaknesses:**

- Missing information on human study. Detailed information on human study could be provided to assess the reliability of the outcome. How do you recruit and select them based on what qualification? Isn't there any conflict of interest for the subjects and authors? How many subjects are recruited? What was the confidence of the votes? This issue is the major reason for leaning toward rejection. I am eager to see the author's feedback for reassessment.

- This framework relies on recently proposed models like a strong LLM, CLIP, and an image composition model. The collection of previous works provides shallow techniques compared with them.

- A missing related work. DenseDiffusion [1] may be worth being included in the layout-to-image generation subsection in Sec. 2 and the box-level multi-modal guidance in Sec. 3.4. DenseDiffusion tried to manipulate attentional weights to control the regions for layout guidance selectively. It would be appreciated if you could compare your method with it for readers in the upcoming revised manuscript. Note that, due to the narrow accessibility to this work at the time of submission, this is not considered in the score evaluation.

[1] Kim, Y. et al. (2023). Dense Text-to-Image Generation with Attention Modulation. http://arxiv.org/abs/2308.12964

**Questions:**

- Minors:
  - In Sec. 3.1, Models. -> Models (Please exclude the period in the section title.)
  - In Fig. 2, it would be inappropriate to include the logo of a commercial product (ChatGPT from OpenAI) in an academic paper. And the company may not allow the usage of their logo to promote the work.

**Details Of Ethics Concerns:**

I leave the comment that "In Fig. 2, it would be inappropriate to include the logo of a commercial product (ChatGPT from OpenAI) in an academic paper. And the company may not allow the usage of their logo to promote the work." for the possible legal issue.

---

> ### Author Response · Authors · 2023-11-23
> **Response to Reviewer kt1C**
>
> We thank reviewer for providing insightful comments. Please find our responses addressing the specific queries below.
>
> **1. Missing information on human study.**
>
> Our approach is designed to overcome the limitations observed in diffusion models when confronted with lengthy and intricate textual prompts that involve multiple objects. Given the absence of an efficient metric tailored for this specific challenge, we conducted a thorough human study to offer a more comprehensive evaluation, as outlined in the main paper. The subjects for human study were recruited from a pool of academic students with backgrounds in computer science and engineering.
> To ensure an unbiased evaluation and maintain objectivity, the subjects participating in the evaluation remained anonymous with no conflicts of interest with the authors.  Furthermore, we prioritized user confidentiality by refraining from collecting personal details such as names, addresses etc.
>
> The entire survey process was anonymized, ensuring that there was no leakage of data to any user. Following a 2-AFC (two-alternative forced choice) design, each participant was tasked with selecting one image out of a pair of images, which best aligns with the given textual prompt and faithfully captures all the objects. Each pair of images was randomly sampled from a pool containing images from both baselines as well as our approach. This system mitigates biases which can originate from user-curated surveys. Further, the pool of images was shuffled before sampling each time. The system was further equipped with the feature of randomly flipping the position of the images, to remove the user-specific position bias in case any pair of images is repeated. Additionally, a 2-second delay between questions was introduced to facilitate more thoughtful decision-making.
>
> The system yielded approximately 350 responses from 50 subjects, with each user answering an average of 7 questions. The results, as presented in Fig. 4 of the main paper, indicate that users predominantly selected images generated from our approach. This observation is further supported by qualitative analyses (Fig. 5 of the main paper and Figs. 9 and 10 of the appendix) and quantitative comparisons in Section 4 of the main paper and Table 1 of the revised appendix. We believe that this comprehensive approach and the subsequent results contribute significantly to the robustness and reliability of our outcomes.
>
>
> **2. This framework relies on recently proposed models like a strong LLM, CLIP, and an image composition model. The collection of previous works provides shallow techniques compared with them.**
>
> As per our understanding of the question, our methodology leverages the complementary strengths of diverse existing approaches, enhancing and refining each component in the process. Furthermore, our approach incorporates specific modifications to optimize the performance of each individual element. To prevent the Large Language Model (LLM) from generating results that lack coherence, we implement an interpolation mechanism. Additionally, we address potential issues in the output of the first-stage model through an iterative refinement scheme. In the context of the composition model, we introduce multi-modal guidance. The the impact of these components is visually demonstrated in Figures 6 and 7 of the main paper. Thus, our proposed layout generation, iterative refinement and multi-modal guidance solves for the weaknesses of the existing components and enhances the overall generalization towards complex textual prompts.
>
>
> **3. Comparison with DenseDiffusion**
> Please refer to Fig. 10 of revised appendix for qualitative comparisons with DenseDiffusion [1]. Additionally, refer to the Table 4 below  for quantitative comparisons. Our method compares favorably well against DenseDiffusion both qualitatively and quantitatively (in terms of PAR score).
>
> Table4. Quantitative comparison with DenseDiffusion in terms of proposed PAR score.
>
> | Method|PAR score |
> |-----|-----|
> | DenseDiffusion | 52\% |
> | Ours  | 85\%  |
>
>
> **4. Use of chatGPT logo.**
>
> We understand the reviewer's concerns. Thank you for pointing it out. We have removed the ChaTGPT logo in the new revised version of the paper. We have further removed the period from Sec. 3.1 title.
>
> [1] Kim, Y. et al. (2023). Dense Text-to-Image Generation with Attention Modulation. http://arxiv.org/abs/2308.12964

---

### Official Review · Reviewer_Br9r · 2023-10-28

**Soundness:** 3 good
**Presentation:** 2 fair
**Contribution:** 2 fair
**Rating:** 5
**Confidence:** 4

**Summary:**

This work aims to solve the problem that prior text-to-image models cannot accurately follow the object specifications in lengthy prompts. The work proposes a novel two-step pipeline that first uses a scene blueprint, which is an LLM-generated layout with object descriptions, to generate the overall image. A CLIP-based guidance is then applied to perform iterative refinement in order to make sure the content of each box is correct. The method enables accurate and diverse image generation with intricate and lengthy input text prompts. The user study and qualitative comparison indicate a non-trivial improvement over baseline methods.

**Strengths:**

1. In addition to previous methods using the layouts for initial image generation, the method further proposes box-based refinement to improve the ability to generate all the objects mentioned in a lengthy and intricate prompt.
2. The method points out the fact that there currently lacks a pipeline for benchmarking text-to-image methods with lengthy prompts and proposes a metric called prompt adherence recall (PAR) to evaluate their method and several baselines.
3. Their method has better prompt adherence compared to baselines in prompt generation. The user study also confirms that the method can faithfully generate objects in the prompt.

**Weaknesses:**

1. This work builds on LLM-grounded Diffusion, as mentioned in Sec 3.3. However, the difference between LLM-grounded Diffusion and the current work is not clearly explained. This work lists "scene blueprints using LLMs" as one of the contributions, but both LLM-grounded Diffusion and LayoutGPT (with both works cited in related work section) propose using LLM to generate the scene layouts with object descriptions. The authors show that the scene representation enhanced by the proposed method works better for long text prompts, but this claim of "proposing a structured scene representation" is not a contribution as it has been proposed and used in previous works.
2. The interpolation of k generated layouts could potentially cause degenerate or undesired results. For example, for "a cat and a dog, one on each side", the LLM can generate 1) a cat on the left, a dog on the right, or alternatively 2) reverse the position for the cat and the dog. The interpolation of layout 1) and 2) results in two objects placed both at the center.
3. Missing details: In Sec 3.3, the authors mention "image-to-image translation" with latent/stable diffusion that leads to generation with higher quality, but no details such as the prompts or the exact approach are given. Is this simply adding some noise and denoise?
4. The evaluation of the work does not investigate whether the proposed method degrades the fidelity of the image. The user study only involves asking whether objects are present and layouts are accurately generated rather than comparing the overall image quality. The author needs to present evaluation results that show their method does not have significant degradations on quality. If only the presence of objects is considered, a simple "copy-and-paste" from baseline text-to-image model with individual object generation using the generated layout would also result in high object presence without bringing much utility to the research community and potential applications.

A typo that does not affect the rating:
1. api -> API (page 7)

**Questions:**

The authors are encouraged to respond to and address the weaknesses in the section above.

---

### Official Review · Reviewer_c4Qg · 2023-11-06

**Soundness:** 3 good
**Presentation:** 3 good
**Contribution:** 3 good
**Rating:** 6
**Confidence:** 4

**Summary:**

This paper presents a novel approach for improving text-to-image generation in diffusion-based models when processing complex scenes with multiple objects and intricate text prompts. The authors leverage Large Language Models (LLMs) to extract the layout information, detailed text descriptions, and background information from text prompts. Then the proposed layout-to-image generation model is composed of two stages: Global Scene Generation and Iterative Refinement Scheme. The Global Scene Generation phase uses object layouts and background context to create an initial scene which roughly represents the target image layout but not very accurate. Then the Iterative Refinement Scheme iteratively evaluates and refines box-level content to align them with their textual descriptions and recompose objects to ensure consistency. Extensive experiments are conducted to validate the effectiveness of the proposed approach.

**Strengths:**

1. The proposed approach can handle the complex scenarios of text-to-image generation with long text prompts very well.
2. The paper is well-written and easy to follow.
3. The iterative refinement loop provides a possible solution to generating images of complex scenes.

**Weaknesses:**

1. I think the major limitation of the proposed approach is efficiency. The complexity of the proposed approach increases as the number of objects increases for complex scenes. This might also be the major issue presenting this approach to be applied in real usage.
2. It seems that although the generation of objects can be iteratively refined, the bounding box locations cannot be refined. If the LLM predicts unreasonable bounding box layouts at the first stage, it cannot be corrected. Have the authors think of introducing the refinement of bounding box locations into the pipeline?
3. How many layouts are needed to interpolate? How does the number of layouts for interpolation affect the results?

**Questions:**

Please refer to the weakness section.

Post-rebuttal: I have read the author response and other reviewers' comments. I decide to keep my initial rating unchanged, but I won't fight for acceptance of this paper.

---

> ### Author Response · Authors · 2023-11-23
> **Response to Reviewer c4Qg**
>
> We thank reviewer for providing insightful comments. Please find our responses addressing the specific queries below.
>
> **1. On efficiency of the proposed approach.**
>
> Our approach operates as a two-stage framework, relying significantly on the efficacy of the underlying models it employs. While some existing state-of-the-art methods exhibit faster performance, they encounter challenges in faithfully capturing every detail of lengthy and intricate textual prompts. Notably, these methods often fall short in generating all the objects specified in the prompt, as illustrated qualitatively in (Figs. 1 and 5) of main paper and (Figs. 9 and 10) of appendix. Our approach demonstrates a high abjectness score expressed as Prompt Adherence Recall (PAR) rate, indicating a notable alignment between the number of objects mentioned in the prompt and those actually present in the generated image (please see Sec. 4 in the main paper). Therefore, a discernible trade-off emerges between addressing complexity and ensuring the faithful reproduction of images. We provide a comparison of our method with existing approaches in terms of time efficiency and PAR score below.
>
> Table 2. Quantitative comparison of our approach with baselines in terms of PAR score and inference times. Kindly note that all inference times are measured on a single Nvidia A100 GPU.
>
> | Method|PAR score |Inference Time (min) |
> |-----|-----|---|
> | Stable diffusion | 49\%  | 0.18 |
> | GLLIGEN |  57\% | 0.5 |
> | LayoutGPT | 69\%   | 0.83 |
> | DeepFloyd |  60\% | 8.33 |
> | DenseDiffusion | 52\%  | 2.5|
> | Ours  | 85\%  | 3.16  |
>
> Recent works, such as those introduced by [1] and [2], have presented methods capable of scaling up diffusion inferences. Looking ahead, we anticipate incorporating these advancements in speed-boosting techniques into our ongoing research as a future work.
>
>
> **2. Unlike objects, bounding box locations cannot be refined. If the LLM predicts unreasonable bounding box layouts at the first stage, it cannot be corrected. Have the authors think of introducing the refinement of bounding box locations into the pipeline?**
>
> Our method is based on two stage process of layout generation and bounding box refinement. The spatial faithfulness of final generated image depends on the quality of layouts generated in the first stage.  We observe that ChatGPT is effective in generating faithful layouts  (please refer to Sec. A.7 of appendix) which exactly follow the details from the textual prompt. [3] further verifies this finding. However, in extreme cases, such as in Fig. 6 of the main paper and Fig. 12 of the revised appendix where object positions are not explicitly defined in the prompt, ChatGPT produces bounding boxes that do not coherently align with the prompts. We tackle such cases by generating multiple layouts, selecting the relevant ones via clustering and then interpolating them to a single optimal layout to lessen the influence of any outlier bounding boxes. We further provide a user-specific hyperparameter $\eta$ (refer to Fig. 3 of main paper) which can be controlled to vary the location of bounding boxes. Refinement of bounding boxes along with object refinement is an interesting future research idea, but is likely to further scale up the compute cost which may not be ideal for user experience.
>
> **3. How many layouts are needed to interpolate? How does the number of layouts for interpolation affect the results?**
>
> While there is no restriction on the number of layouts needed for interpolation, our final results are generated with the interpolation of 3 layouts. Please refer to Figures 11 and 12 (Sec A.6) of the revised appendix for a visualization of generated images with different number of layouts. We conclude that while interpolation plays a significant role on the final generated image in extreme cases where spatial locations are not clear from the prompt, it does not have a drastic impact on the final generated image when object locations are well-defined.
>
>
> [1] Sylvain Gugger et al. "Accelerate: Training and inference at scale made simple, efficient and adaptable". https://github.com/huggingface/accelerate
>
> [2] Zhang, Kexun, et al. "ReDi: Efficient Learning-Free Diffusion Inference via Trajectory Retrieval." ICML 2023.
>
> [3]  Lian, L., Li, B., Yala, A., & Darrell, T. (2023). LLM-grounded Diffusion: Enhancing Prompt Understanding of Text-to-Image Diffusion Models with Large Language Models. arXiv preprint arXiv:2305.13655.

---

### Official Review · Reviewer_XXwY · 2023-11-06

**Soundness:** 3 good
**Presentation:** 3 good
**Contribution:** 2 fair
**Rating:** 5
**Confidence:** 3

**Summary:**

The paper extends recent works which leverages layouts to generate scenes corresponding to complex text-prompts. This work first shows that for complex prompts, existing layout to image generation methods have certain failure modes and proposes some practical modifications which are augmented with existing layout to scene generation methods.  First, the authors propose a scene blueprint to represent complex text-prompts; Secondly the authors design an iterative refinement process which improves the alignment of generated images with the complex text-prompts.

**Strengths:**

- The research question is a very practical problem — usually most of the text-to-image generation models are not good at coherent images corresponding to complex prompts, so providing a solution for it is important.
- The method consists of various components (a lot of these components are existing though) and is conceptually intuitive!
- The framework obtains strong results on human-study for fidelity of images generated for long prompts.

**Weaknesses:**

Cons / Questions

- While the writing is satisfactory, it can still be improved! The authors should provide more information in the paper on how Eq. (5) is used to guide the sampling process.
- Can the authors provide more intuition on the interpolation step?
- Given that there are stronger open-source diffusion models (e.g., DeepFloyd) — the authors should provide some context on how long prompts work in those cases, as they use a stronger text-encoder like T5.
- While the authors comment that the size of the tokens (77 in CLIP) is one of the potential reasons on why SD cannot generate compositional prompts — I believe this is only partially true. Even for non-complex compositional prompts, SD is not able to generate coherent images. Can the authors comment in general on some potential reasons why these text-to-image models are not able to generate images corresponding to simple compositional or complex prompts? I think both are related somehow and it will be beneficial to provide some context regarding it.

**Questions:**

See Cons/Questions;
Overall, the paper is practical, but the various components though intuitive are not technically novel. While I do agree that not everything in a paper needs to be novel — the authors should provide solid justifications on the design of each component.

I am happy to revisit my scores after the rebuttal!

---

> ### Author Response · Authors · 2023-11-23
> **Response to Reviewer XXwY (1/2)**
>
> We thank reviewer for providing insightful comments. Please find our responses addressing the specific queries below.
>
> **1. Information on how Eq. (5) is used to guide the sampling process and clarity in writing.**
>
> We kindly refer the Reviewer to section A.8 of revised appendix for a detailed mathematical explanation on sampling process using guidance.
>
> We will refine the writing for better clarity and ease of understanding in our final manuscript.
>
> **2. More intuition on the interpolation step.**
>
> We observed that proprietary LLMs such as ChatGPT (used in our approach) are good at following the details from the prompts and generating image blueprint in the form of box proposals as per the prompt. The same has been verified in [1]. However, we observed that in the multi-object settings where the spatial position of objects in unclear, ChatGPT has certain inherent bias for such cases. We conducted an experiment where we instructed ChatGPT to provide boxes for the prompt "In a living room with a dog and a cat sitting one on each side". Note that, in this prompt, the relative position of cat and dog is not clear. We observed that 60\% of the times ChatGPT generated bounding boxes towards the left for the dog and only 30\% times the dog was on the right. For the remaining 10\% of the times, it had arbitrary position.
> To avoid such errors and to account for this bias, we instead prompt ChatGPT to output $K$ bounding boxes per object proposal. To exploit the above bias, we perform density clustering and choose the cluster containing the majority of the boxes. After this, we introduce the interpolation step to merge these filtered bounding boxes into a single optimal box. This potentially removes the effect of the outlier boxes if present. We have further presented a visual demonstration of the effect of such irrelevant boxes on the final generated image in Fig. 6 of the main paper and Fig. 12 of the revised appendix. Kindly note that this is the extreme case where the position of the objects is not clear from the text. So to further facilitate the user-specific control, we provide a hyperparameter $\eta$ which can be controlled to adjust the position of the boxes. Kindly refer to Sec. 3.3 of the main paper for the description of $\eta$ and Fig. 3 (main paper) for qualitative visualization of the effect of $\eta$.
>
>
> **3. Comparison with DeepFloyd.**
>
> As recommended, we compare our method with the DeepFloyd and present visual comparisons in Fig. 10 (section A.5) of the revised appendix. Additionally, we also present quantitative comparisons in terms of Prompt Adherence Recall (PAR) below in Table 1. We observe that even with a strong T5 text encoder, DeepFloyd struggles to generate coherent images with complex compositional prompts. We conclude that our scene blueprint augmented with iterative refinement is effective for generating coherent images from complex compositional prompts.
>
> Table 1. Quantitative comparison of our approach with DeepFloyd.
>
> | Method| PAR score |Inference Time (min)|
> |-----|-----|---|
> | DeepFloyd |  60\% | 8.3 |
> | Ours  | 85\%  | 3.16 |
>
> **4. Potential reasons why these text-to-image models are not able to generate images corresponding to simple compositional or complex prompts?**
>
> We agree with the reviewer's comments that current text-to-image diffusion models struggle to generate the image content even for non-complex compositional prompts. Experiments on DeepFloyd with a strong text encoder show that even stronger models struggle to generate coherent images. As per our intuition, there could be several factors that could contribute to inconsistencies in text-to-image diffusion models to generate images corresponding to simple compositional or complex prompts.
>
> - Lack of generalizability towards diverse textual prompts can lead to images with missing details.
>
> - The limitation could also stem from the lack of compositional prompts in the training dataset of text-to-image diffusion model inhibiting it from generalizing and generating diverse and accurate images for those prompts.
>
> - Lack of location-aware training of diffusion models on complex compositional prompts can limit their ability to capture intricate relationships between objects, scenes, or attributes, leading to incoherent images.
>
>
>
> [1] Lian, L., Li, B., Yala, A., \& Darrell, T. (2023). LLM-grounded Diffusion: Enhancing Prompt Understanding of Text-to-Image Diffusion Models with Large Language Models. arXiv preprint arXiv:2305.13655.

---

> > ### Author Response · Authors · 2023-11-23
> > **Response to Reviewer XXwY (2/2)**
> >
> > **5. Justifications on the design of each component.**
> >
> > Our methodology leverages the complementary strengths of diverse existing approaches, enhancing and refining each component in the process. Furthermore, our approach incorporates specific modifications to optimize the performance of each element. To prevent the Large Language Model (LLM) from generating results that lack coherence, we implement an interpolation mechanism. Additionally, we address potential issues in the output of the first-stage model through an iterative refinement scheme. In the context of the composition model, we introduce multi-modal guidance. The impact of these components is visually demonstrated in Figs. 6 and 7 of the main paper. In this manner, our proposed Scene Blueprint generation, iterative refinement, and multi-modal guidance enhance the overall generalization towards complex textual prompts.

---

### Author Response · Authors · 2023-11-23
**Thank you for the valuable feedback**

We sincerely thank the reviewers (XXwY, c4Qg, Br9r, kt1C) for their detailed and positive feedback. The idea of improving text-to-image diffusion models on complex prompts is a very practical research problem and is conceptually intuitive (XXwY), proposed iterative refinement approach is an effective solution for handling complex scenarios of text-to-image generation, especially with long text prompts (c4Qg and Br9r). The proposed prompt adherence Recall (PAR) addresses the lack of a pipeline for benchmarking text-to-image methods with lengthy prompts (kt1C). The proposed approach ensures high prompt adherence recall and is supported with a user study confirming the faithful generation of objects in the prompt ( Br9r, kt1C). The paper is well-written and easy to follow (c4Qg).


We are pleased to observe the overall engaging comments provided by all reviewers. We have diligently addressed each concern raised by the reviewers individually, and the main PDF has been accordingly updated including a revised appendix attached at the end of paper. Below, we present a summary of our responses.

- **Further Comparisons:** In response to the suggestions from **Reviewers XXwY and kt1C**, we have included qualitative comparisons with DeepFloyd and DenseDiffusion in **Figure 10** and quantitative comparisons in **Table 1** of the revised appendix.

- **Guidance Function:** Addressing the recommendation from **Reviewer XXwY**, we offer additional insights into how the guidance loss is incorporated into the sampling process during the Iterative Refinement phase.

- **Interpolation:** Building upon the suggestions from **Reviewers XXwY and c4Qg**, we provide further intuition about Interpolation and its impact on the final generated image. **Figures 11 and 12** in the revised appendix delve into this aspect. Additionally, responding to input from **Reviewers Br9r and c4Qg**, we present a technical explanation on handling ambiguous prompts with unclear object locations. This is supported by a study on the layout generation capability of ChatGPT, included in detail in **Table 2** of the revised appendix.

- **Justification on underlying components:** In accordance with the recommendations from **Reviewers XXwY, Br9r, and kt1C**, we offer clear and comprehensive distinctions between our approach and the underlying components it utilizes. We provide justifications for each component and highlight the proposed novelties, signifying a substantial advancement in addressing a significant challenge in text-to-image diffusion models.

- **Additional Details:** In response to the suggestion from **Reviewer Br9r**, we have included additional details about the image-to-image translation step used in our work. We have also included these details in **Sec. A.2** of the appendix. Furthermore, as recommended by **Reviewer kt1C**, we have provided extensive details about the user study, encompassing information about the subjects, underlying settings, and procedure.

- **Efficiency:** As suggested by **Reviewer c4Qg**, compelling reasons supporting the superiority of our approach, both qualitatively **(Fig. 3 main paper and Figures 9, 10  appendix)** and quantitatively **(Table 1 in the revised appendix)**, have been included. We also outline future directions for accelerating our method.

- **Study on Fidelity:** Responding to the suggestion from **Reviewer Br9r**, we have conducted another comprehensive evaluation of our approach in terms of fidelity through an extensive user study in **Fig. 13 of revised appendix**, the details of which are included in the individual response.

- **Use of ChatGPT logo:** As recommended by **Reviewer kt1C**, we have removed the ChatGPT logo from the main block diagram of to avoid ethical concerns in the revised paper.


**Our codes will be publicly released. All suggested changes will be reflected in our final manuscript.** We summarize the salient features of our proposed approach below:

- To the best of our knowledge, our work is the first public attempt to enable diffusion models to generate images with complex and detailed textual prompts, ensuring a faithful representation of details.
- We introduce  Scene Blueprints as a structured scene representation encompassing scene layout and object descriptions that enable a coherent step-wise
generation of images from complex and lengthy prompts.
- We propose an Iterative Refinement scheme,  which improves the box proposals corresponding to each object guided by a multi-modal signal in a multi-step fashion. Our refinement scheme helps in generating missing objects in the scene and provides fine-grained control over the objects.
- We present quantitative and qualitative results showcasing the effectiveness of our method, in terms of adherence to textual descriptions, demonstrating its utility and benefit in text-to-image synthesis from lengthy prompts.

---

### Meta-Review · Area_Chair_3Nwh · 2023-12-08

**Metareview:**

This paper proposes an approach to handle complex compositional prompts for text-to-image generative models, in particular by instructing an LLM to generate a bounding box layout based on the query. Evaluation is based on a user study, but also includes qualitative results for ablations, and some quantitative measures based on object detection in the generated images. The reviews appreciate the studied problem and obtained results. The novelty is considered to be in the combination of existing components and experimentation with the approach. The author rebuttal has partially addressed the points raised in the reviews. One reviewer has raised their score to 6 in response to the rebuttal, one reviewer did not respond to the rebuttal. Overall the ratings are borderline, two just above and two just below.

**Justification For Why Not Higher Score:**

The recommendations by reviewers are borderline, which does not warrant a higher recommendation.

**Justification For Why Not Lower Score:**

There are two slightly negative recommendations:
* Reviewer XXwY did not respond to the rebuttal, but it seems there concerns have been addressed.
* Reviewer Br9r maintained a 5 rating after rebuttal, but concerns seem mostly addressed. In their response they refer to Dalle3 tech report to tone done the novelty claim: Dalle3 was released just days before the paper submission deadline, and the tech report is not a peer-reviewed publication. Their point about the bounding box interpolation does not seem a reason for reject. Finally they mention a paper LLM-grounded-diffusion, which was not referred to in the original review.

---

### Decision · Program_Chairs · 2024-01-16

Accept (poster)